# Microbiota of Urine, Glans and Prostate Biopsies in Patients with Prostate Cancer Reveals a Dysbiosis in the Genitourinary System

**DOI:** 10.3390/cancers15051423

**Published:** 2023-02-23

**Authors:** Micael F. M. Gonçalves, Teresa Pina-Vaz, Ângela Rita Fernandes, Isabel M. Miranda, Carlos Martins Silva, Acácio Gonçalves Rodrigues, Carmen Lisboa

**Affiliations:** 1Division of Microbiology, Department of Pathology, Faculty of Medicine, University of Porto, 4200-319 Porto, Portugal; 2Department of Urology, University Hospital Centre of São João, 4200-319 Porto, Portugal; 3Department of Surgery and Physiology, Faculty of Medicine, University of Porto, 4200-319 Porto, Portugal; 4Cardiovascular R&D Centre—UnIC@RISE, Faculty of Medicine, University of Porto, 4200-319 Porto, Portugal; 5Centre for Health Technology and Services Research/Rede de Investigação em Saúde (CINTESIS/RISE), Faculty of Medicine, University of Porto, 4200-319 Porto, Portugal; 6Department of Dermatology and Venereology, University Hospital Centre of São João, 4200-319 Porto, Portugal

**Keywords:** prostate cancer, microbiota, urinary tract, glans penis, urine

## Abstract

**Simple Summary:**

Prostate cancer (PCa) is the most common cancer diagnosed among men aged 50 years and older. There is gaining interest in the genitourinary microbiome in developing ways to control this disease. In this work, an analysis of the bacterial microbiota was performed from urine samples, glans secretions, and prostate biopsies from patients with PCa, and the results were investigated and compared with non-PCa patients. Our results showed a distinct clustering of genera associated with urine samples and prostate biopsies of PCa and non-PCa patients. Observed microbial dysbiosis may increase chronic inflammation and ultimately prostate carcinogenesis. Future research on the biological functions of the uropathogens found is needed to understand their impact on the pathogenesis of PCa.

**Abstract:**

Prostate cancer (PCa) is the most common malignant neoplasm with the highest worldwide incidence in men aged 50 years and older. Emerging evidence suggests that the microbial dysbiosis may promote chronic inflammation linked to the development of PCa. Therefore, this study aims to compare the microbiota composition and diversity in urine, glans swabs, and prostate biopsies between men with PCa and non-PCa men. Microbial communities profiling was assessed through 16S rRNA sequencing. The results indicated that α-diversity (number and abundance of genera) was lower in prostate and glans, and higher in urine from patients with PCa, compared to non-PCa patients. The different genera of the bacterial community found in urine was significantly different in PCa patients compared to non-PCa patients, but they did not differ in glans and prostate. Moreover, comparing the bacterial communities present in the three different samples, urine and glans show a similar genus composition. Linear discriminant analysis (LDA) effect size (LEfSe) analysis revealed significantly higher levels of the genera *Streptococcus*, *Prevotella*, *Peptoniphilus*, *Negativicoccus*, *Actinomyces*, *Propionimicrobium*, and *Facklamia* in urine of PCa patients, whereas *Methylobacterium*/*Methylorubrum*, *Faecalibacterium*, and *Blautia* were more abundant in the non-PCa patients. In glans, the genus *Stenotrophomonas* was enriched in PCa subjects, while *Peptococcus* was more abundant in non-PCa subjects. In prostate, *Alishewanella*, *Paracoccus*, *Klebsiella*, and *Rothia* were the overrepresented genera in the PCa group, while *Actinomyces*, *Parabacteroides*, *Muribaculaceae* sp., and *Prevotella* were overrepresented in the non-PCa group. These findings provide a strong background for the development of potential biomarkers with clinical interest.

## 1. Introduction

According to the International Agency for Research on Cancer of the World Health Organization, prostate cancer (PCa) is the most common cancer diagnosed among men aged 50 years and older and the fifth leading cause of cancer-associated death [1]. In 2020, approximately 473,000 new cases of PCa and 108,000 deaths were estimated in Europe. Based on the reports, the prevalence of this neoplasm is increasing worldwide [2]. New cases of PCa in Portugal are estimated to be approximating 6700, and there have been 1900 cancer-associated deaths, whose circumstances nowadays pose an important and worrying public health problem [1].

In the past decades, PCa has aroused the attention of scientists due to its high mortality rates. Unfortunately, the etiology and pathology of PCa remain unknown [3]. However, it has been reported that PCa may be caused by genetic mutations and external risk factors, including environmental factors, such as dietary changes and lifestyle, microbial infections, and inflammatory stimuli [4,5,6]. Moreover, PCa treatment significantly affects men’s sexual health, reducing their quality of life [7].

Notably, all the above-mentioned factors are known to affect the composition of the resident microbiota in the skin, mucous membranes, fluids, and tissues of the human body [8]. Recent studies on the human microbiome have shown that mammary, lung, bladder, pancreas, intestine, and prostate microbial dysbiosis can enhance the progression of diseases, such as cancer and chronic inflammation [9,10,11,12,13,14,15,16].

The genitourinary microbiota is gaining a relevant role in the prostate carcinogenesis process: emerging evidence suggests that the resident microbiota may promote chronic prostate inflammation linked to the development of PCa [6,17]. Any imbalance in the composition of the genitourinary microbiota can lead to an increase in immune responses or a modification in the extracellular environment of the prostate [18]. Although it is well established that *Helicobacter pylori* affects gastric physiology which can lead, at last instance, to the development of gastric cancer [19], the relationship between specific pathogens and PCa is still poorly understood. Nevertheless, it has been reported that some pathogenic microorganisms may possibly induce symptomatic and asymptomatic infections in the prostate, including the bacterium *Cutibacterium acnes* (syn. *Propionibacterium acnes*) [20], and sexually transmitted infection-causing pathogens, such *as Chlamydia trachomatis* [21], *Neisseria gonorrhoeae* [22], *Trichomonas vaginalis* [23], and *Mycoplasma genitalium* [24].

Most of the above findings were obtained by performing conventional polymerase chain reaction (PCR), quantitative real-time PCR techniques, and Sanger sequencing technology. Meanwhile, some studies have applied metagenomic sequencing technologies for an accurate and complete characterization of the genitourinary microbiota, and its possible implications in the pathogenesis of PCa. The current studies carried out concerning the genitourinary microbiota are mainly focused on the role of the urinary [25,26], fecal [25,27,28], rectal [29], prostate [18,30,31,32], and prostatic fluid [33] microbiota in men with PCa. Taken together, these results strongly support the hypothesis that the microbiota might be involved in prostate carcinogenesis, its progression, and relapse.

Until now, a detailed and comprehensive analysis that combines the microbiota in the urine, glans, and prostate biopsies of PCa and non-PCa patients has not been conducted. Therefore, the aims of our study were to characterize and compare the potential association between the urinary, glans, and prostate microbiotas of PCa and non-PCa men. This research contributes to improve our understanding of the role of the genitourinary microbiota in PCa risk and pathogenesis, as well as to elucidate potential biomarkers for diagnosis and eventually new therapeutic and prognostic options.

## 2. Materials and Methods

### 2.1. Study Population and Clinical Data

This study was conducted under the project “SexHealth & Prostate Cancer, Psychobiological Determinants of Sexual Health in Men with Prostate Cancer”, approved by the Ethical Committee of the University Hospital Center of São João (Portugal). From the patients enrolled in this project with clinical suspicion of PCa admitted to the Department of Urology between February 2022 and August 2022, 30 males (15 positive and 15 negative cases for PCa) were randomly chosen for this study. Prostate cancer cases were histologically confirmed as prostate adenocarcinoma by the hospital. Negative cases for PCa were considered as the non-PCa group. Non-PCa patients were asymptomatic patients (no urogenital symptoms) referred to the Department of Urology due to elevated PSA. Patients did not report to have LUTS (lower urinary tract symptoms). Patients with other urinary pathologies that could introduce bias were excluded from this study.

All participants signed a written informed consent to contribute their own anonymous information to this study. Age, clinical and analytical data such as PSA value, prostate volume evaluated by magnetic resonance imaging (MRI), Charlson’s index, and ISUP grade were recorded. All participants started antibiotic prophylaxis the day before the biopsy procedure according to the local protocol.

### 2.2. Specimen Collection

Urine, glans swabs, and prostate biopsy specimens were collected from each male. Urine specimens were maintained at room temperature (less than 3 h after collection) and immediately processed in the laboratory. Glans secretions swabs and prostate biopsy tissues were maintained at 4 °C until further processing. First-void urine stream specimens were collected by the clean catch method, then centrifuged at 9000 × rpm for 15 min, and the pellets were immediately stored at −80 °C. Sterile swabs moistened with physiological saline solution were used for the collection of glans secretions, transferred to Eppendorf tubes with 1 mL of phosphate-buffered saline solution (PBS) and then vortexed for 1 min. Afterwards, the swabs were discarded, and the samples were centrifuged at 14,000 × rpm for 5 min and the pellets stored at −80 °C. Prostate biopsies were performed under transrectal ultrasound image, with 12 random cores including variable cores to a specific target if the MRI identified one or the Urologist found it adequate. Then, one random core was transferred to tubes with 1 mL of TripleXtractor solution (GRiSP, Porto, Portugal), and stored at −80 °C until DNA extraction.

### 2.3. DNA Extraction

All specimens’ samples from PCA and non-PCa groups were maintained at room temperature (15–25 °C) for 2 min before DNA extraction. Prostate biopsy samples (*n* = 30) were cut into small pieces and transferred to tubes with 200 µL of tissue lysis buffer solution (Qiagen, Hilden, Germany). Urine (*n* = 30) and glans swab (*n* = 30) pellets were resuspended in PBS. Total genomic DNA was extracted using the QIAamp DNA Micro Kit (Qiagen, Hilden, Germany), following the manufacturer’s instructions. The quality of the DNA was assessed by agarose gel electrophoresis (0.8%). DNA purity and quantity were determined using a Qubit 2.0 fluorometer and Qubit dsDNA BR kit (Thermo Fisher Scientific, Waltham, MA, USA).

### 2.4. Illumina High-Throughput Sequencing

The 16S rRNA gene microbiome profiling with Illumina MiSeq platform was performed by STAB Vida (Lisbon, Portugal) by amplifying the hypervariable V3–V4 region with primers 341F (5′ CCTACGGGNGGCWGCAG 3′) and 785R (5′ GACTACHVGGGTATCTAATCC 3′). The resulting data were analyzed according to STAB Vida standard protocols using QIIME2 v2021.4 [34]. The reads were denoised using the DADA2 plugin [35] as follows: trimming and truncating low quality regions, quality filtering (to remove reads with less than 300 bp, ambiguous bases (“N”), and sequences with an average quality lower than Q30), dereplicating the reads, and identification and removal of chimeric reads. High-quality reads were assigned to operational taxonomic units (OTUs), and then classified by taxon using a fitted classifier. The scikit-learn classifier was used to train the classifier using the SILVA database (release 138 QIIME) [36], with a dynamic clustering threshold of 99% similarity. For classification purposes, only OTUs containing at least 10 sequence reads were considered as significant. Given the nature of the samples and the low microbial biomass, an additional step, attempting to remove contaminant OTUs based on the prevalence and frequency of the determined OTUs, was applied. This was achieved using the Bioconductor’s decontam package [37].

### 2.5. Bioinformatic and Statistical Analysis

The marker-gene data profiling module of the MicrobiomeAnalyst web-based platform [38] was used for community profiling and comparative analysis. The parameters of the low count filter (minimum count of 4 and 20% prevalence) and low variance filter (10% based on the interquartile range) were used as default. The data were normalized to rarefy the data to the minimum library size, scaled by total sum scaling, but we did not apply any data transformations. The α- and β-diversity metrics were calculated based on OTUs abundance table at the genus level. Number of observed OTUs (richness index) and diversity (Shannon–Wiener index) of bacteria taxonomic diversity were calculated as a measure of α-diversity. The distance between samples based on the difference in OTUs in each sample defined as β-diversity, was evaluated by the principal coordinates analysis (PCoA) and hierarchical clustering analysis using a Bray–Curtis dissimilarity index.

Statistical analyses were performed using SPSS Statistics v26 (IBM Corporation, Endicott, NY, USA). Before the analysis, the data were checked for normality using the Shapiro–Wilk test. As the data met the analysis of variance (ANOVA) assumptions, one-way ANOVA followed by a *t*-test was used to determine differences between the two groups based on clinical characteristics and the significance of the taxonomy. Permutational multivariate analysis of variance (PERMANOVA), based on 999 permutations, was used to test for significant differences in sampling units between the PCa and non-PCa groups. This was performed separately for the urine, glans swabs, and prostate biopsy samples. A *p*-value less than 0.05 was considered statistically significant. OTUs at the genus level with a relative abundance of at least 0.01% and present in more than 60% of tested samples were used to calculate the core community at the genus level in each sample type and in each group. Linear discriminant analysis (LDA) effect size (LEfSe) was adopted to explore the significant differences in bacterial taxa abundance [39]. The LEfSe submodule in the MicrobiomeAnalyst platform was used with the default settings of an FDR-adjusted, Kruskal–Wallis *p*-value cutoff set to 0.1 and the logarithmic LDA score cut-off at 2.0.

## 3. Results

### 3.1. Clinical Characteristics of Participants

The clinical information of the 30 Caucasian men who participated in this study is presented in Table 1. Among them, 15 men were included in the PCa group (68 ± 9 years of age), and the other 15 men were included in the non-PCa group (69 ± 8 years of age). Both groups showed similar mean age and comparable comorbidities, as seen by Charlon’s index. PSA levels and prostate volumes were higher in the non-PCa group, since most of them have benign prostatic hyperplasia (BPH).

### 3.2. Sequencing Characteristics

A total of 5,654,185 raw sequence reads were obtained with an average of 62,824 ± 47,688 (average ± SD) reads per sample. A total of 3,946,884 effective reads were generated, and each sample produced an average of 45,366 ± 33,686 effective reads. Rarefaction curves are listed in Appendix A. After denoising, a total of 3486 unique features (OTUs) were identified.

### 3.3. Bacterial Community Structure and Composition

#### 3.3.1. α- and β-Diversity

Boxplots of α-diversity in terms of urine, glans swabs, and prostate biopsies microbiota richness and diversity for PCa and non-PCa groups are shown in Figure 1A,B. When comparing the three biological samples, urine samples were the only ones to show low indices of species richness and diversity. Although no significant differences (*p* ≥ 0.05) were observed between PCa and non-PCa groups, the species richness was lower in prostate biopsies and glans swabs, and higher in urine from patients with PCa. Additionally, the bacterial diversity (Shannon–Wiener index) was lower in prostate biopsies and higher in urine and glans swabs from patients with PCa.

The PCoA and hierarchical clustering analysis used to compare the similarity between the PCa and non-PCa groups in each sample type and all combined samples are shown in Figure 2. The results indicated that the composition of the bacterial community was significantly different in urine samples between the PCa and non-PCa groups (PERMANOVA, R^2^ = 0.068, *p* = 0.014) (Figure 2A), while in swabs of glans and prostate biopsy samples the composition of the bacterial community did not differ (PERMANOVA, R^2^ = 0.020, *p* = 0.946; R^2^ = 0.026, *p* = 0.689, respectively) (Figure 2B,C).

The PCoA analysis revealed that the bacterial communities present in urine and glans swabs of both patient groups are similar to each other but distinct from the bacterial communities of the prostate (Figure 2D). Moreover, this observation is also notorious in the hierarchical cluster, where it is clear that urine and glans samples of both patient groups were grouped into a large cluster.

#### 3.3.2. Taxonomic Composition of Bacterial Communities

Following initial evaluations for differences in overall bacterial community composition, we investigated the relative abundance of specific taxa between the PCa and non-PCa groups. A total of seven different phyla, eleven classes, twenty-six orders, thirty-four families, and fifty genera were determined across all samples. Assessment of the relative abundance of the top 25 bacteria was considered as the predominant bacteria at each taxon level (Figure 3). At the phylum level, *Firmicutes*, *Proteobacteria*, *Actinobacteriota*, *Bacteroidota*, *Campilobacterota*, *Cyanobacteria*, and *Fusobacteriota* were the dominant bacterial phyla in both PCa and non-PCa patients (Figure 3A) in each sample type. Among them, *Firmicutes* was significantly more abundant in urine samples of PCa patients (0.469%) than in non-PCa (0.212%), *p* = 0.014, while *Proteobacteria* was significantly more abundant in urine of non-PCa patients (0.338%), compared to those with PCa (0.083%), *p* = 0.014 (Appendix A).

At the class level, the dominant bacterial classes from both groups were Clostridia, Actinobacteria, Gammaproteobacteria, Bacteroidia, Bacilli, Negativicutes, Campylobacteria, Alphaproteobacteria, Cyanobacteria, Fusobacteria, and Coriobacteria (Figure 3B). Among them, Clostridia was significantly more abundant in urine samples of PCa patients (0.229%) than in non-PCa (0.078%), *p* = 0.023, while Gammaproteobacteria in non-PCa patients was significantly more abundant (0.320%) than in PCa patients (0.070%), *p* = 0.014 (Appendix A).

Among the top 10 genera more abundant, higher levels of *Prevotella*, *Corynebacterium*, *Finegoldia*, *Peptoniphilus*, *Fenollaria*, *Streptococcus*, *Negativicoccus*, *Enterococcus*, *Porphyromonas*, and *Ezakiella* were observed in the urine of PCa patients, compared to non-PCa patients (Figure 4A). On the contrary, *Klebsiella* was significantly more abundant in non-PCa patients (*p* = 0.043) (Appendix A). Although no statistically significant differences were found, urine samples from the non-PCa group tended to show a higher abundance of the genera *Staphylococcus* and *Escherichia*/*Shigella* (Figure 4A).

Similar compositions of bacterial abundance were observed in both the PCa and non-PCa groups regarding the glans samples (Figure 4B). The ten dominant bacterial genera were *Prevotella*, *Corynebacterium*, *Finegoldia*, *Porphyromonas*, *Campylobacter*, *Staphylococcus*, *Peptoniphilus*, *Fenollaria*, *Anaerococcus*, and *Mobiluncus* (Figure 4B). Only *Actinomyces* was found to be more abundant in PCa patients (*p* = 0.041).

Regarding prostate biopsies, similar bacterial composition was also observed in PCa and non-PCa patients but with differences in their relative abundances. The PCa group tended to show more abundance of the genera *Pseudomonas*, *Faecalibacterium*, *Cutibacterium, Bacteroides*, *Corynebacterium*, *Turicella*, *Curvibacter*, *Sphingomonas*, and *Staphylococcus* (Figure 4C). *Cutibacterium* (*p* = 0.014) and *Lawsonella* (*p* = 0.005) were statistically different. On the other hand, the genus *Prevotella* was revealed to have the higher mean relative abundance in non-PCa patients with statistically significant differences (*p* = 0.027).

Differences in the bacterial communities between both groups and biological samples are also presented in a heatmap with all genera listed (Appendix A).

#### 3.3.3. Bacterial Core Community

The bacterial core community at the genus level with relative abundance of at least 0.01% and present in more than 60% of samples is shown in Figure 5. The most prevalent core OTUs in urine samples of PCa patients correspond to the genus *Prevotella* followed by *Streptococcus*, *Corynebacterium*, and *Finegoldia*, while in non-PCa patients, *Corynebacterium* and a not-assigned genus were the most prevalent (Figure 5A).

The most dominant genera in glans were similar between PCa and non-PCa patients. *Prevotella* was the most prevalent genus in both groups, followed by *Corynebacterium*, *Peptoniphilus*, *Finegoldia*, and *Anaerococcus* in patients with PCa, while in non-PCa was followed by *Finegoldia*, *Porphyromonas*, *Peptoniphylus*, *Corynebacterium*, and *Anaerococcus* (Figure 5B). Regarding prostate biopsies, a higher prevalence of a not-assigned genus, followed by *Pseudomonas*, *Cutibacterium*, *Curvibacter*, *Sphingomonas*, *Corynebacterium*, *Staphylococcus*, *Lawsonella*, and *Paracoccus* were observed in PCa patients. On the other hand, *Pseudomonas* was the most prevalent genus in non-PCa patients, followed by a not-assigned genus, *Sphingomonas*, *Curvibacter*, *Corynebacterium*, and *Cutibacterium* (Figure 5C).

#### 3.3.4. Differential Abundance of Bacterial Taxa

The linear discrimination analysis (LDA) effect size (LEfSe) method was employed to search differentially abundant genera within the PCa and non-PCa groups (Figure 6A–C).

This analysis revealed that the genera *Streptococcus*, *Prevotella*, *Peptoniphilus*, *Negativicoccus*, *Actinomyces*, *Propionimicrobium* (*P. lymphophilum*), and *Facklamia* were significantly increased in urine samples of PCa subjects, whereas *Methylobacterium*/*Methylorubrum*, *Faecalibacterium*, *Blautia*, and one not-assigned group were increased in non-PCa patients (Figure 6A). In the case of glans samples from PCa subjects, the genus *Stenotrophomonas* was enriched, while *Peptococcus* was underrepresented (Figure 6B). Analysis of the prostate biopsies’ microbiota also revealed *Alishewanella*, *Paracoccus*, *Klebsiella*, and *Rothia* as the most overrepresented genera in PCa subjects, while *Actinomyces*, *Parabacteroides*, *Prevotella*, and members of the family *Muribaculaceae* were enriched in non-PCa subjects (Figure 6C).

LEfSe analysis for the top 30 significantly enriched genera, combining the urine, glans swabs, and prostate biopsies from subjects with PCa and non-PCa is shown in Figure 6D. This analysis showed which taxa are most abundant in each group. For example, *Enhydrobacter* and *Anaerococcus* were more abundant in the three types of samples in patients with PCa compared to patients with non-PCa. Nevertheless, the differences in the *Anaerococcus* abundance between both PCa and non-PCa groups were more evident in urine samples. Moreover, higher abundances of *Peptoniphilus* and *Actinomyces* in urine and swabs of patients with PCa was also evident. Regarding non-PCa patients, the genera *Blautia* and *Faecalabacterium* were more abundant in the three types of samples compared with patients with PCa. However, the abundance of these genera was clearer in urine and swabs samples.

## 4. Discussion

Microbiome composition and its function have been identified as contributing factors in the pathogenesis of several diseases [40]. An imbalance in microbiome homeostasis owing to persistent inflammation induced by the resident microbiota may be involved in the development of malignant diseases, such as cancer [41,42,43].

However, little is known regarding the differences in the genitourinary tract microecology in PCa and non-PCa patients. To the best of our knowledge, this study represents the first detailed and comprehensive analysis of the genitourinary tract bacterial microbiota of urine, glans, and prostate biopsies between PCa and non-PCa patients.

The findings from this study show disparities in the structure and bacterial composition of urine, glans, and prostate biopsies between PCa and non-PCa patients. Nevertheless, these differences in the bacterial community are more evident in urine samples. Alpha diversity metrics revealed that bacterial diversity was lower in prostate biopsies from patients with PCa compared to non-PCa patients. In fact, Ma et al. [33] demonstrated that PCa subjects exhibited a reduced microbial diversity in the prostate when compared to the non-PCa subjects; they suggested that the bacterial diversity might have a role in the development of PCa. According to our results, the urine of patients from both groups were the only samples to show low indices of species richness and diversity. Such an outcome suggests that dysbiosis of the urine microbiota may be the origin of prostate inflammation [44]. 

Moreover, the similarity found in the bacterial community in the urine and glans, compared to that of the prostate, might be explained by their anatomic proximity.

Various studies reported the genera *Prevotella*, *Corynebacterium*, *Streptococcus*, *Finegoldia*, *Peptoniphilus*, *Anaerococcus*, *Propionibacterium*, *Staphylococcus*, and *Lactobacillus* in the urinary microbiota of adult men [45] as well as in the male genital mucosa [46]. These genera have been considered as commensals of the genitourinary system [47,48,49]. In agreement to these previous reports, our results also demonstrated that *Prevotella*, *Corynebacterium, Staphylococcus, Peptoniphilus,* and *Anaerococcus* are among the top 10 genera in urine and glans samples of both PCa and non-PCa patients.

Regarding PCa patients, LEfSe analysis revealed that *Streptococcus*, *Prevotella*, *Peptoniphilus*, *Negativicoccus*, *Actinomyces*, *Propionimicrobium*, and *Facklamia* were enriched in urine. Some of these genera, that were differentially abundant in PCa patients, harbor uropathogens that colonize the genital tract, including *Propionimicrobium lymphophilum* and *Streptococcus anginosus* that have been implicated in urogenital infections, including prostatitis [26,50,51]. Furthermore, anaerobic cocci, such as *Peptoniphilus* and *Negativicoccus*, have been reported in urinary tract microbiota and in cases of urinary tract infections, mostly in patients with comorbidities [52,53]. A previous study by Hurst et al. [54] identified also *Propionimicrobium*, *Negativicoccus*, *Peptoniphilus*, and *Prevotella* in the urine of patients with PCa. Garbas et al. [3] suggested that the presence of these uropathogens might be implicated in prostate inflammation progression.

The prostate microbiota in patients from both the PCa and non-PCa groups was dominated by the genera *Pseudomonas*, *Cutibacterium*, *Curvibacter*, *Sphingomonas*, and *Corynebacterium*, although their abundances were more evident in patients with PCa. On the other hand, *Pseudomonas* was the most prevalent in non-PCa patients. Feng et al. [32] observed that the genus *Pseudomonas* was prevalent in prostatic cancer tissues along with a greater expression of human small RNAs in patients with low rates of metastases. The authors concluded that *Pseudomonas* infection hampers the progression to metastatic disease and might be negatively correlated with tumor–node–metastasis (TNM) stage. Our results also showed that *Stenotrophomonas* was significantly elevated in glans samples of PCa patients. The presence of this genus that often co-colonizes with *Pseudomonas*, is also found to be negatively correlated with TNM, thus hindering the progression of metastatic cancer [55]. However, in-depth studies need to corroborate this hypothesis.

*Cutibacterium* is a pro-inflammatory bacterial genus that has been detected in prostatic tissues and has been widely studied in the genitourinary context [42,56]. Specifically, *Cutibacterium acnes* has been proposed to induce the expression of immunosuppressive genes in macrophages, which in turn influence the risk of PCa progression [57,58]. Additionally, Yu et al. [59] reported an abundance of the genus *Sphingomonas* in prostatic secretions of patients with PCa, while Yow et al. [60] observed that *Curvibacter* and *Corynebacterium* were the most abundant genera within aggressive PCa tissues. In our study, apart from the above-mentioned genera, we observed that *Staphylococcus*, *Lawsonella,* and *Paracoccus* were also prevalent in the prostate biopsies of PCa patients. The presence of the genus *Staphylococcus* in prostate tissues is corroborated by a study carried out by Cavarretta et al. [18]. Furthermore, Sarkar et al. [61] investigated the differential composition of commensal bacteria in prostate tissues among patients with BPH and PCa, and observed *Prevotella copri*, *Cupriavidus campinensis*, and *C. acnes* were the most abundant bacteria in diseased prostate tissues of PC patients. To our knowledge, there is no evidence of the association between *Lawsonella* and PCa; it has only been described in association with breast cancer and infection [62,63,64].

Fastidious Gram-positive taxa such as *Actinomyces* spp. have been recognized recently as etiological agents of urogenital infections, such as urethritis, cystitis, and prostatitis [65,66,67,68]. *Actinomyces* causes a slowly progressive infection with tissue destruction that often resembles malignancy. In the LEfSe analysis (Figure 6D), our results showed that *Actinomyces* was more abundant in the three biological samples of patients with PCa.

Other genera were observed to be significantly elevated in prostatic tissues of patients with PCa, such as *Alishewanella*, *Paracoccus*, *Klebsiella*, *Rothia*, and *Microbacterium*. On the contrary, Sarkar et al. [61] observed that the genus *Paracoccus* was enriched in the prostate tissue of patients with BPH. Further investigations regarding the association of these genera with PCa are needed to confirm these results. Nevertheless, using a meta-transcriptomic approach, Salachan et al. [12] highlighted *Microbacterium* sp. as an interesting candidate for further investigation due to its association with PCa, as they revealed significantly higher abundances of *Microbacterium* sp. in samples from advanced PCa.

Considering the bacterial abundance at the genus level, a higher number of bacterial genera were found to be overrepresented in the urine and biopsies from PCa subjects in comparison with the non-PCa ones, as shown in out LEfSe analysis. Therefore, these results strongly suggest the occurrence of alterations in the genitourinary system, which can induce a chronic inflammatory environment in the prostate and, ultimately, the development of cancer as previously advocated by Garbas et al. [3] and Shrestha et al. [26]. Similar to other age-related diseases, PCa is often characterized by enhanced oxidative stress and oxidative damage. Recent findings demonstrate the link between the enhanced indices of oxidative stress and PCa [69], and between radical PCa removal and the normalization of such indices revealed by measuring 8-OHdG and 8-Iso-PGF2α in the urine of patients with PCa [70]. Moreover, the measurement of such biomarkers in the urine before and after surgery helps to predict radicality (and perhaps local recurrence) following surgery [70].

It is worth mentioning that the classic sexually transmitted bacteria, such as *Chlamydia trachomatis* and *Neisseria gonorrhoeae*, were not identified in this study cohort. Nevertheless, recent studies reported that sexually transmitted microorganisms might be involved in prostate carcinogenesis [21,22,23,24].

Our study has some limitations. First, we did not test for possible sources of contamination during medical procedures as all analyses were based on existing total DNA. Additionally, despite the analysis of three sample types of each subject, the number of patients could be greater, and the lack of true control samples limits our conclusions between PCa and non-PCa groups. Finally, taking an antibiotic the day before the biopsy might influence our findings, but both groups of patients underwent this antibiotic prophylaxis which does not affect the differences found in the genitourinary microbiota.

The detection of microorganisms in the urine or prostate opens a new and exciting field for science. Until recently, the urine was considered a sterile niche in the human body, and its normal flow prevents bacteria from infecting the urinary tract [71]. With the development of next-generation sequencing technologies, alterations in the urine or prostate microbiome (or even in adjacent sites) associated with PCa have increasingly become the focus of current research. However, the overall number of studies conducted in this field is still limited and difficult to compare the results obtained. One important factor that strongly influences the comparison of microbiome data from different studies is the method of collecting and processing the biological samples. To date, studies to determine the prostate microbiome of patients with PCa or BPH have analyzed both formalin-fixed paraffin-embedded tissue samples [18,30] and fresh material [55]. Another aspect is the region of the prostate where the biopsy is performed and how urine is collected. By analyzing a random fragment of the prostate and owing to its small size, it is difficult to control the distribution of microorganisms in the organ or determine the most colonized area, as highlighted by Alexeyev et al. [42]. Thus, standardization of the collection technique should be mandatory for further microbiome studies.

## 5. Conclusions

In conclusion, our results show that there is microbial dysbiosis with an increase in overall species diversity in the urine of patients with PCa compared to non-PCa patients. This dysbiosis may promote chronic inflammation in the prostate and might be implicated in the development of PCa. Differential abundances of certain bacterial genera present in the urine, glans, and prostate could provide together with data from other studies, promising biomarkers for early diagnosis, and aid the scientific and clinical community to search for new therapeutic and prognostic options. Furthermore, this study did not prove any connection between bacterial sexually transmitted pathogens and PCa.

The emerging human pathogens that colonize the genital tract should be carefully investigated once they have been implicated in urogenital infections. Such studies would provide a starting point for future investigations into the functional relevance of these uropathogens in PCa pathogenesis. These aspects should be further validated in future research, including a larger cohort, and other molecular and histological tools.

## Figures and Tables

**Figure 1 cancers-15-01423-f001:**
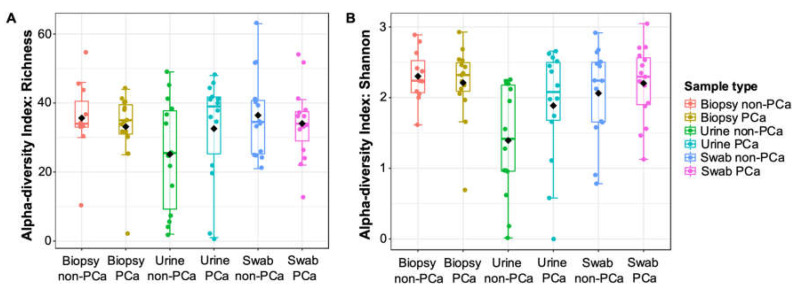
The α-diversity metrics (**A**) richness and (**B**) Shannon index for PCa and non-PCa groups by sample type. One-way analysis of variance (ANOVA) followed by *t*-test was used. No significant differences (*p* > 0.05) were observed.

**Figure 2 cancers-15-01423-f002:**
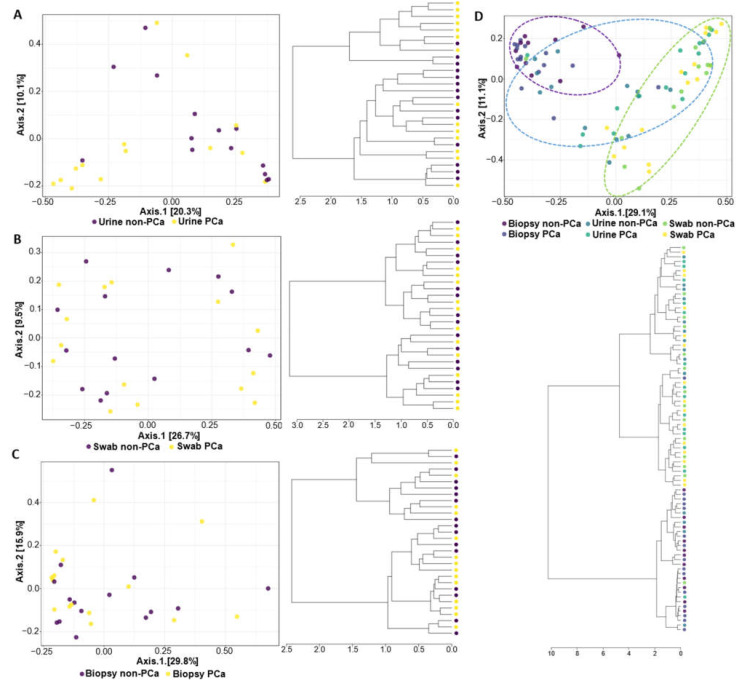
The β-diversity metrics (PCoA on the top and hierarchical cluster on the bottom) for PCa and non-PCa groups were categorized by (**A**) urine, (**B**) glans, (**C**) prostate biopsies, and (**D**) all combined samples. The blue, green, and purple ellipses represent the urine, glans, and biopsy samples, respectively.

**Figure 3 cancers-15-01423-f003:**
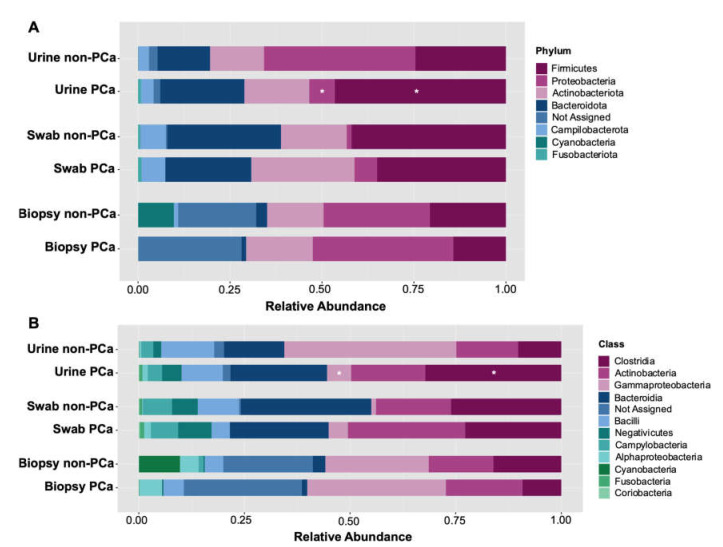
Relative abundance of (**A**) bacterial phyla and (**B**) classes in each sample type for PCa and non-PCa groups. Asterisks indicate significant differences between both groups (*t*-test was used; *p* < 0.05).

**Figure 4 cancers-15-01423-f004:**
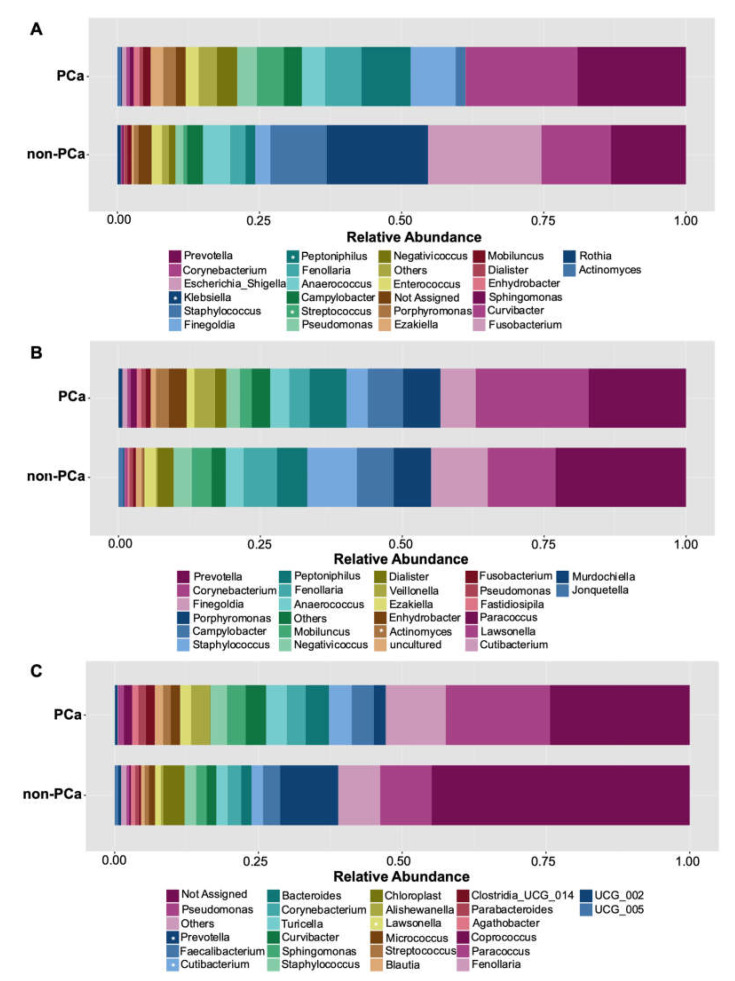
Relative abundance of the 25 most abundant genera in (**A**) urine, (**B**) glans, and (**C**) prostate biopsies for PCa and non-PCa groups. Asterisks indicate significant differences between both groups (*t*-test was used; *p* < 0.05).

**Figure 5 cancers-15-01423-f005:**
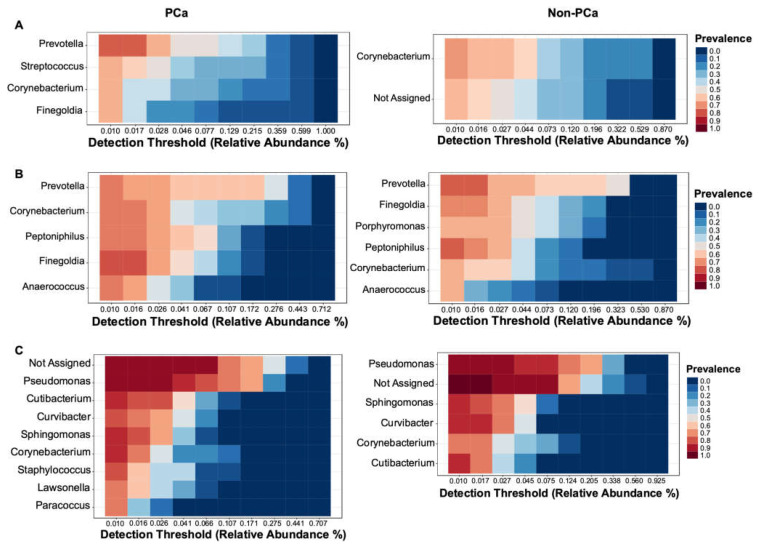
Core community analysis at genus level detected in more than 60% of the samples of (**A**) urine, (**B**) glans, and (**C**) prostate biopsies for both PCa (left) and non-PCa (right) patients.

**Figure 6 cancers-15-01423-f006:**
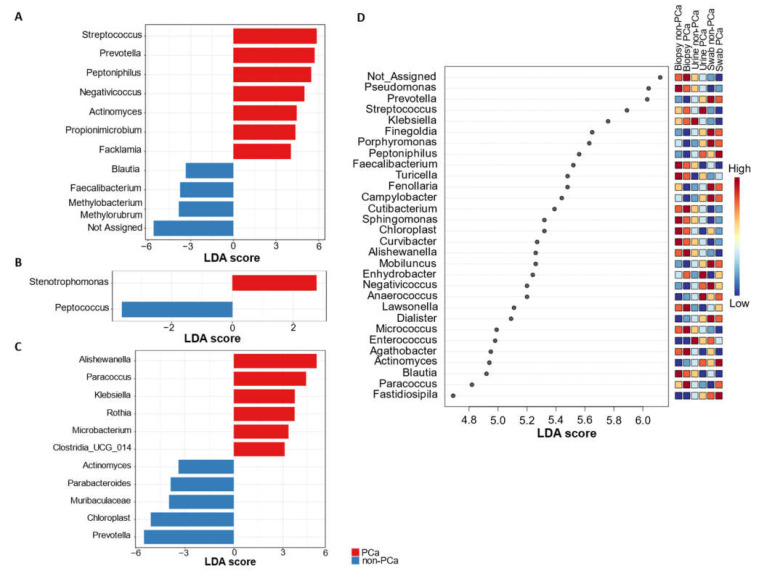
LefSe analysis of bacterial genus abundance for PCa and non-PCa groups categorized by (**A**) urine, (**B**) glans, (**C**) prostate biopsies, and (**D**) all combined samples.

**Table 1 cancers-15-01423-t001:** Clinical characteristics of men enrolled in the present study. PCa: prostate cancer; PSA: prostate-specific antigen; ISUP: International Society of Urological Pathology; SD: standard deviation; nd: not defined.

Characteristics		PCa Group	Non-PCa Group	*p*-Value
Subjects		15	15	nd
Age, years: mean (SD)		68 (9)	69 (8)	0.621
PSA, ng/mL: median (range)		9 (3–28)	9 (3–32)	0.640
Prostate volume, g: mean (range)		37 (56)	79 (230)	0.015
Charlson index: *n* (%)	1	2 (13%)	2 (13%)	0.799
	2	4 (27%)	2 (13%)	
	3	4 (27%)	5 (33%)	
	4	1 (7%)	3 (20%)	
	5	2 (13%)	2 (13%)	
	6	1 (7%)	1 (7%)	
	7	1 (7%)	0	
ISUP grade: *n* (%)	1	3 (20%)	nd	nd
	2	6 (40%)	nd	
	3	2 (13%)	nd	
	4	2 (13%)	nd	
	5	2 (13%)	nd	

## Data Availability

Data can be accessed from corresponding author upon request.

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
