# Peer review of "Microbiota of Urine, Glans and Prostate Biopsies in Patients with Prostate Cancer Reveals a Dysbiosis in the Genitourinary System"

_cancers, 2023, doi:10.3390/cancers15051423_

Round 1

Reviewer 1 Report

In this manuscript, the authors aimed to evaluate the association of microbiome and prostate cancer and to reveal the potential link of microbiome in prostate carcinogenesis. They enrolled 30 suspicious men and collected urine, glan and prostate biopsy samples for DNA extraction and then 16s rRNA sequencing and data analysis. Some genera were found differentially between PCa and non-PCa. However, this is a study with relatively small sample size, and there are more room to improve the quality of the work for publication. 

1. The authors mentioned 30 suspicious men enrolled. Surprisingly, they found the exactly equal number of PCa and non-PCa. Please comment it, and clarify how the authors design the study. 

2. In table 1, the p values should be presented. Second, please comment the level of PSA in PCa is lower than non-PCa. Third, for the range, it should be minimum to max. 

3. why urine specimen were kept at room temperature before processing? how long were they kept at room temperature? would it affect the change of microbiome?

4. Figure 1 legend, p value for no significant difference should be >0.05?

5. Figure 2D, it would be better to draw circles to show the difference between the tissue profile from urine/glan profile of microbiome

6. Figure 5, the authors claim differences in relative abundance of some genera. It is expected to provide p values. In addition, it would be better to present the data in heatmap to show the differences between PCa vs. non-PCa. 

7. The authors might consider to leave the second aim (prostate carcinogenesis) out. 

8. The authors may think about some more literature citation in the introduction to support some statements (Poore GD, Nature, 2020;579:567-74, PMID: 32214244; Zhu G, Eur J Cancer; 2021;151:25-34; PMID: 33962358; Hernandez BY, Cancer Epidemiol Biomarkers Prev, 2022;31:221-9; PMID;34697061), so that readers may read the literature for relevant statements. 

Author Response

Reviewer 1

In this manuscript, the authors aimed to evaluate the association of microbiome and prostate cancer and to reveal the potential link of microbiome in prostate carcinogenesis. They enrolled 30 suspicious men and collected urine, glan and prostate biopsy samples for DNA extraction and then 16s rRNA sequencing and data analysis. Some genera were found differentially between PCa and non-PCa. However, this is a study with relatively small sample size, and there are more room to improve the quality of the work for publication.

Reply: We agree that the number of patients could be higher; nevertheless, our results involved 90 sequenced samples in total. In this study we used three sample types of each subject, representing the first detailed analysis of the genitourinary tract bacterial microbiota (urine, glans, prostate), and even so showing differences between PCa and non-PCa patients. As we mentioned in conclusion section this study provides a starting point for future investigations into the functional relevance of uropathogens in the PCa pathogenesis.

1. The authors mentioned 30 suspicious men enrolled. Surprisingly, they found the exactly equal number of PCa and non-PCa. Please comment it, and clarify how the authors design the study.

Reply: This study was conducted under the ongoing project “SexHealth & Prostate Cancer, Psychobiological Determinants of Sexual Health in Men with Prostate Cancer”, that aims to study the impact of sexual behavior and sexual attitudes of patients with PCa in engaging in risky sexual behaviors, and the potential effect of infections caused by some of the organisms most often transmitted sexually. In this way, one of the tasks was the study of the genitourinary microbiome of these patients. Therefore, from the total patients enrolled in this project with clinical suspicion of PCa admitted to the Hospital 30 males (15 positive and 15 negative cases for PCa) were randomly chosen for this study. This was clarified in section 2.1.

2. In table 1, the p values should be presented. Second, please comment the level of PSA in PCa is lower than non-PCa. Third, for the range, it should be minimum to max.

Reply: Table 1 has been revised and p values are now included. We recalculated the medians, and they are similar between groups, with both having a median 9 (p value 0.640). There are no significant differences. The range with minimum to maximum was also included.

3. why urine specimen were kept at room temperature before processing? how long were they kept at room temperature? would it affect the change of microbiome?

Reply: Urine samples were maintained at room temperature (less than 3 hours after collection) and then immediately processed in the laboratory to not affect the microbiome. We added this in the manuscript (L112-L113). This step was followed according to Shrestha et al. (2018) (10.1016/j.juro.2017.08.001).

4. Figure 1 legend, p value for no significant difference should be >0.05?

Reply: Right. Sorry, it was a mistake. Changed.

5. Figure 2D, it would be better to draw circles to show the difference between the tissue profile from urine/glan profile of microbiome

Reply: We agree and changed the figure.

6. Figure 5, the authors claim differences in relative abundance of some genera. It is expected to provide p values. In addition, it would be better to present the data in heatmap to show the differences between PCa vs. non-PCa.

Reply: The sentence was revised to make it clearer. The heatmap for all genera represented in the urine, glans, and prostate biopsies samples for both groups is available in Figure S2.

7. The authors might consider to leave the second aim (prostate carcinogenesis) out.

Reply: We agree with the reviewer. We excluded the second aim regarding the etiology of prostate carcinogenesis.

8. The authors may think about some more literature citation in the introduction to support some statements (Poore GD, Nature, 2020;579:567-74, PMID: 32214244; Zhu G, Eur J Cancer; 2021;151:25-34; PMID: 33962358; Hernandez BY, Cancer Epidemiol Biomarkers Prev, 2022;31:221-9; PMID;34697061), so that readers may read the literature for relevant statements.

Reply: Thanks! We added in the introduction.

Reviewer 2 Report

The paper submitted by Gonçalves et al represents  the first detailed and comprehensive analysis of the genitourinary tract bacterial microbiota of urine, glans, and prostate biopsies between the patients with or without prostate cancer. The biggest issue is that there is no information about the patients without prostate cancer (non-PCa). Do the non-PCa patients also have urinary system diseases or not? If those non-PCa patients also have urinary system diseases, it may caused the comparison of the two groups have no solid evidence to show the real differences on phylum, class or genus levels. There are also some other questions rosed up to me when I read through the paper:

1.     There is no significant difference between the PCa and non-PCa. so the description from Line 199-202 are not rigorous.

2.     Line 259-260: Are these genera significantly more abundance with statistical analysis?

3.     Line 331-335: Does the cancer reduced bacterial diversity, or the reduced bacterial diversity caused cancer? I want to hear the opinion from the authors. Thanks!

4.     Line 431-432: there is no significant difference between PCa and non-PCA for prostate biopsy samples statistically, if I understand correctly. Thus, it is not rigorous to describe the reduced overall species diversity in the prostate.

Author Response

Reviewer 2

The paper submitted by Gonçalves et al represents the first detailed and comprehensive analysis of the genitourinary tract bacterial microbiota of urine, glans, and prostate biopsies between the patients with or without prostate cancer. The biggest issue is that there is no information about the patients without prostate cancer (non-PCa). Do the non-PCa patients also have urinary system diseases or not? If those non-PCa patients also have urinary system diseases, it may caused the comparison of the two groups have no solid evidence to show the real differences on phylum, class or genus levels. There are also some other questions rosed up to me when I read through the paper:

Reply: Non-PCa patients were asymptomatic patients (no urogenital symptoms) referred to the Department of Urology due to elevated PSA. Patients did not report to have LUTS (Lower Urinary Tract Symptoms). Patients with other urinary pathologies that could introduce biases were excluded from this study. This was clarified in section 2.1.

1. There is no significant difference between the PCa and non-PCa. so the description from Line 199-202 are not rigorous.

Reply: We understood the reviewer’s concern and are aware that there are no significant differences between both groups in α-diversity. However, we just meant to describe the tendency observed between PCa and non-PCa groups. As suggested, the sentence was revised to make it clearer.

2. Line 259-260: Are these genera significantly more abundance with statistical analysis?

Reply: Of the 25 most abundant genera in PCa, Cutibacterium and Lawsonella were statistical different. We added this information with P-value in the text.

3. Line 331-335: Does the cancer reduced bacterial diversity, or the reduced bacterial diversity caused cancer? I want to hear the opinion from the authors. Thanks!

Reply: We understand that the sentence might not be clear. At our knowledge, the low microbial diversity and changes in microbial composition might be associated with a greater risk for the development of cancer. Such fact is corroborated by Ma et al., who showed a reduced bacterial diversity in the cancer group, compared with the non-cancer group. Considering our results, we also demonstrated that a dysbiosis in urine microbiota may be indicative for the development of prostate inflammation. This has been made clearer in the revised version of the manuscript.

4. Line 431-432: there is no significant difference between PCa and non-PCA for prostate biopsy samples statistically, if I understand correctly. Thus, it is not rigorous to describe the reduced overall species diversity in the prostate.

Reply: We agree with you and we changed the sentence accordingly.

Reviewer 3 Report

The study aims to characterize and compare the potential association between urinary, glans, and prostate microbiotas of PCa and non-PCa men; and to assess their relevance in terms of prostate carcinogenesis. The manuscript is well-written and interesting. Minor revision is required.

- what kind of biopsies did you perform? How may cores? please provide those data.

- how did you collect prostate volume? TRUS, MRI? 

- As already reported in the manuscript, similar to other age-related diseases, PCa is often characterized by enhanced oxidative stress and oxidative damage. For this scope, recent findings demonstrate the link between the enhanced indices of oxidative stress and PCa; and between radical prostate cancer removal and the normalization of such indices revealed by measuring 8-OHdG and 8-Iso-PGF2α in the urine of patients with prostate cancer. Moreover, the measurement of 8-OHdG and of 8-Iso-PGF2αin in urine before and after surgery as a technique to help predict radicality (and perhaps local recurrence) following surgery (DOI: 10.3390/jcm11206102). I strongly believe this should be included in your paper.

- Summarise the conclusions

- Please 

- check typos

Author Response

Reviewer 3

The study aims to characterize and compare the potential association between urinary, glans, and prostate microbiotas of PCa and non-PCa men; and to assess their relevance in terms of prostate carcinogenesis. The manuscript is well-written and interesting. Minor revision is required.

- what kind of biopsies did you perform? How many cores? please provide those data.

Reply: Prostate biopsies were performed using transrectal ultrasound, with 12 random cores including variable cores to a specific target if the MRI identified one or the Urologist found it adequate. This was clarified in section 2.2.

- how did you collect prostate volume? TRUS, MRI?

Reply: Prostate volumes were measured using MRI.

- As already reported in the manuscript, similar to other age-related diseases, PCa is often characterized by enhanced oxidative stress and oxidative damage. For this scope, recent findings demonstrate the link between the enhanced indices of oxidative stress and PCa; and between radical prostate cancer removal and the normalization of such indices revealed by measuring 8-OHdG and 8-Iso-PGF2α in the urine of patients with prostate cancer. Moreover, the measurement of 8-OHdG and of 8-Iso-PGF2αin in urine before and after surgery as a technique to help predict radicality (and perhaps local recurrence) following surgery (DOI: 10.3390/jcm11206102). I strongly believe this should be included in your paper.

Reply: We thank the helpful hint and this information was included in the manuscript.

- Summarise the conclusions

- check typos

Reply: The conclusions were summarized as suggested.